# Kalman-Based Scene Flow Estimation for Point Cloud Densification and 3D Object Detection in Dynamic Scenes

**DOI:** 10.3390/s24030916

**Published:** 2024-01-31

**Authors:** Junzhe Ding, Jin Zhang, Luqin Ye, Cheng Wu

**Affiliations:** School of Rail Transportation, Soochow University, Suzhou 215500, China; jzding@stu.suda.edu.cn (J.D.); lqye@suda.edu.cn (L.Y.)

**Keywords:** scene flow estimation, point cloud densification, 3D object detection, Kalman filter

## Abstract

Point cloud densification is essential for understanding the 3D environment. It provides crucial structural and semantic information for downstream tasks such as 3D object detection and tracking. However, existing registration-based methods struggle with dynamic targets due to the incompleteness and deformation of point clouds. To address this challenge, we propose a Kalman-based scene flow estimation method for point cloud densification and 3D object detection in dynamic scenes. Our method effectively tackles the issue of localization errors in scene flow estimation and enhances the accuracy and precision of shape completion. Specifically, we introduce a Kalman filter to correct the dynamic target’s position while estimating long sequence scene flow. This approach helps eliminate the cumulative localization error during the scene flow estimation process. Extended experiments on the KITTI 3D tracking dataset demonstrate that our method significantly improves the performance of LiDAR-only detectors, achieving superior results compared to the baselines.

## 1. Introduction

High-precision environment sensing is essential in advancing autonomous driving technology. It necessitates the integration of multiple sensors such as cameras, millimeter-wave radar, and LiDAR [1]. LiDAR utilizes a restricted set of scanning beams to perceive the environment, leading to the generation of a sparsely populated point cloud. The inadequate semantic information provided by a sparsely populated point cloud significantly impacts the efficacy of 3D perception tasks, including 3D object detection, scene semantic segmentation [2], and 3D object tracking [3]. Point cloud densification is crucial in comprehending 3D environments as it offers supplementary structural and semantic information to improve the accuracy of 3D perception tasks. Nevertheless, existing densification methods based on registration prove inadequate for dynamic targets due to the inherent limitations of the point cloud, such as incompleteness and deformation. To tackle this concern, we propose a Kalman-based scene flow estimation method for point cloud densification and 3D object detection in dynamic scenes.

Generally speaking, point cloud densification aims to reconstruct a dense and precise point cloud frame from a given sequence of sparse point clouds. This process is particularly challenging due to the variability and complexity of spatial data in real-world environments. Many registration-based methods [4,5,6,7,8] are currently being investigated for aligning multi-frame point clouds to obtain dense point cloud maps. The Iterative Closest Point (ICP) algorithm [9] is a classical point cloud alignment algorithm that optimizes the correspondence between two point clouds by minimizing their differences. However, its reliance on static geometries makes it less effective in scenarios where objects or the environment change over time. While the ICP family of algorithms is efficient at aligning large-scale point clouds, it often overlooks the dynamic attributes of the objects within the scene. These dynamic attributes include aspects such as the movement, rotation, and deformation of objects over time, which are crucial in dynamic environments. With the emergence of deep learning, several works [10,11,12,13,14] have started to exploit feature correspondences and spatial relationships within point clouds.

Advancements in feature matching techniques, such as SSL-Net’s approach to sparse semantic learning [15] and PGFNet’s strategy for preference-guided filtering [16], have been instrumental in enhancing the accuracy of feature correspondence in complex scenarios. These methods demonstrate the importance of accurately identifying and filtering key features, a concept that can significantly augment traditional methods like ICP in point cloud densification processes, especially in dynamic scenes. The ability to discern and track these dynamic attributes accurately is crucial for understanding changes in the scene over time, thereby enhancing 3D perception in applications like autonomous driving.

Scene flow estimation aims to fundamentally understand the evolving dynamics of the environment without making any assumptions about the scene structure or object motion. Early works [17,18,19,20] focused on stereo vision cameras, where 3D motion fields were estimated using a knowledge-driven approach. Flownet3D [21] is the first deep learning-based end-to-end scene flow estimation network. It utilizes PointNet++ [22] as the backbone network while introducing a novel stream embedding layer. NSFP [23] is the first method for estimating scene flow based on runtime optimization. It represents the scene flow using a unique implicit regularizer and is not constrained by data-driven limitations. However, when estimating long sequence point cloud scene flow, the estimator is limited by the motion relationships between two frames of point clouds, which can lead to the point cloud being guided to the wrong location. To overcome this localization error, an effective correction mechanism is necessary.

With the development of multi-modality techniques, several studies have proposed utilizing additional information to improve the performance of 3D object detectors. The authors of [7] propose a framework based on an improved ICP method to enhance the performance of deep learning-based classification by incorporating the shape information of the LiDAR point cloud. The authors of [24] explore a motion detection method based on point cloud registration. This method detects motion by analyzing the overlapping relationship between the registered source and target point clouds. The authors of [25] propose a point cloud feature fusion module based on scene flow estimation to enhance the performance of 3D object tracking.

The method proposed in this paper is based on scene flow estimation and aims to improve the density of dynamic objects in a long sequence of point clouds to enhance the performance of 3D object detection. This will allow multiple frames of these objects to converge into the same frame. However, existing scene flow estimators suffer from localization errors when estimating a long sequence of point clouds. These errors disrupt the structural information of the target and render it unsuitable for 3D perception tasks. Therefore, we introduce a Kalman filter to correct the scene flow estimation results and eliminate the localization errors frame by frame. This approach leads to improved performance in point cloud densification. Our contributions in this study are as follows:We propose a novel method for densifying point clouds by utilizing motion information from multi-frame point clouds with a novel scene flow estimator using a two-branch pyramid network as the implicit regularizer.We address the issue of localization error in long-sequence scene flow estimation by utilizing a Kalman filter to precisely adjust the position of dynamic objects.We compare our point cloud densification method with the ICP-based point cloud densification method, and our method outperforms in terms of accuracy and precision.We conduct extensive experiments on the KITTI 3D tracking dataset to demonstrate that our method effectively improves the LiDAR-only detectors’ performance by achieving superior results to the baselines.

## 2. Related Work

### 2.1. Point Cloud Alignment and Densification

Point cloud alignment and densification are vital for 3D environment perception. Initially, methods like the Iterative Closest Point (ICP) and Normal Distributions Transform (NDT) were predominantly used for mapping and densification tasks [26]. Advancements in this field have led to the development of methods that combine learning-based approaches with traditional algorithms, for instance, a hybrid method that integrates learning with ICP for aligning both rigid and variable point clouds, where ‘rigid’ refers to objects that maintain their shape and size over time, while ‘variable’ objects may change shape or deform. This method has shown state-of-the-art performance, whose results are currently among the best in this field of research [27].

Further innovations include the development of low-latency alignment methods suitable for real-world applications, which combine adaptive thresholding ICP, robustness kernels, motion compensation, and downsampling strategies [28]. In dynamic environments, enhanced NDT algorithms have been employed to estimate the static likelihood of points, thereby efficiently handling scenes with multiple moving objects [29]. Urban scene point cloud registration has also seen significant advancements with the introduction of MLP(Multi-layer Perceptron)-based models, which estimate transformations implicitly and show promising results [14]. Large-scale outdoor road scenes have been addressed by selecting key area laser point clouds for feature extraction and registration, focusing on both coarse and fine alignment [30].

The increasing availability of point cloud data has spurred research in scene flow estimation. Large-scale datasets with dense 3D motion ground truth have been developed for depth estimation refinement and scene flow analysis [31]. Innovative self-supervised methods using neural network-based constraints have been proposed to estimate scene flow, enabling multi-frame point cloud densification [23]. Recent advances in feature matching, such as those demonstrated in SSL-Net, PGFNet, JRA-Net, and MSA-Net, have further enhanced the capability to identify reliable correspondences crucial for accurate scene flow estimation [16,32,33]. However, suggestions for further optimization in this area have been made, advocating for the integration of registration, segmentation, and rigid-body flow estimation to enhance densification outcomes [34]. Techniques for fusing scene flow in non-rigid objects, especially in autopilot scenarios, have also been developed to improve the reliability of point cloud densification [35]. Moreover, various studies have employed advanced encoding-decoding architectures for point cloud completion and densification [36,37,38].

### 2.2. 3D Object Detection

The introduction of PointNet [39] and PointNet++ [22] revolutionized the extraction of features from sparse point cloud data, enabling effective classification, detection, and segmentation. Subsequent 3D detectors, trained solely on point cloud data, have shown impressive capabilities. PointPillars, for example, encodes longitudinal pillar point cloud features as pseudo-images for 2D detection in a 3D context [40]. VoxelNet partitions point clouds into fixed-resolution voxels for feature encoding and then uses an RPN (Region Proposal Network) network to perform 3D object detection. However, this representation loses the spatial resolution and geometric features of point cloud and increases the density of voxels, resulting in a cubic multiplication of computational complexity [41].

Recent developments have seen the emergence of self-integrated single-stage detectors for outdoor 3D object detection. By using knowledge distillation, these models optimize performance without additional computational effort [42]. Transformation-Equivariant 3D Detectors have been proposed, employing sparse convolutional backbones to extract transformed equivariant voxel features for high-performance detection [43].

In the PMPF approach for 3D object detection, point cloud data are enhanced via projection onto the image plane and the integration of ‘region pixels’ from corresponding image regions. These region pixels enrich the point cloud with additional image-derived information, allowing for improved performance when used with LiDAR-only detectors. This method demonstrates significant performance improvements in various LiDAR-based detectors [44]. Motion detection methods based on point cloud registration, combining geometric and structural features with neural network-based interests, have enhanced the detection and tracking of moving objects [24]. Novel multi-task models have been introduced for simultaneous scene flow estimation and object detection, achieving significant improvements in performance and latency [45]. These models represent a crucial advancement in the field, combining two complex tasks into a single efficient process.

Overall, the field of point cloud processing, particularly in alignment, densification, and 3D object detection, has seen rapid advancements with the integration of deep learning techniques and innovative algorithmic approaches. These developments not only enhance the accuracy and efficiency of existing methods but also open new avenues for research and application in various domains such as autonomous driving, robotics, and urban planning.

## 3. Proposed Method

In this paper, a two-branch pyramid network is modeled as the scene flow estimator, and the adopted dataset is the KITTI tracking dataset [46]. Scene flow estimation is first performed on pairs of point cloud frames to obtain an optimal estimate of the motion field and store the corresponding optimal model weights; then, these saved optimal model weights are utilized to perform a long sequence of scene flow estimation. In this process, the Kalman filter is employed to rectify the position of dynamic targets obtained from estimation, thereby mitigating the cumulative localization error. The results after correction are refined according to the point cloud density of the current frame to streamline the number of point clouds and minimize the impact of cumulative noise on the results. Finally, the aforementioned operations are recursively applied to the sequence of point clouds from the initial frame to the final frame, resulting in the densification of the dynamic point cloud targets of the dataset. The overall architecture of the densification method is depicted in Figure 1. The pseudocode is presented in Algorithm 1.
**Algorithm 1** Dynamic Point Cloud Densification Algorithm**Require:** Point cloud Sequence: P={P0,P1,P2,…,PN}**Ensure:** Densification Results: D={D1,D2,…,DN}
  1:Initiate the optimal scene flow parameters Θ={θ0→1,θ1→2,…,θN−1→N}  2:**function** *F*(Pi−1, Pi, θi−1→i)  3:      Define the model to calculate the scene flow from Pi−1 to Pi  4:      **return** estimated scene flow  5:**end function**  6:**for** i=1 to *N* **do**  7:      SF0→i=SF0→i−1+F(Pi−1,Pi,θi−1→i), where SF0→i is the scene flow from P0 to Pi  8:      Perform Kalman Filter to SF0→i to eliminate the cumulative localization error  9:      Overlap SF0→i with Pi and perform point cloud refinement10:      Obtain densified result Di11:**end for**12:**return** Densification Results: D={D1,D2,…,DN}


### 3.1. Long Sequence Scene Flow Estimation

Scene flow estimation is the process of determining the motion vectors that describe the dynamic changes between consecutive point cloud frames in a scene. These vectors provide insights into the relative movement of objects and surfaces from one frame to the next.

Let $P_{t}$ and $P_{t+1}$ denote the point cloud frames at consecutive time steps *t* and t+1. The scene flow $SF_{t\to t+1}$ is the vector field that represents the estimated movement of each point from $P_{t}$ to its new position in $P_{t+1}$.

The scene flow $SF_{t\to t+1}$ quantifies the displacement of points between consecutive frames in a point cloud sequence, capturing the dynamic changes within the scene from time *t* to t+1. This concept extends beyond the straightforward difference between coordinates, encompassing complex movements, including the translation, rotation, and deformation of objects in the environment.

In practice, there is not always a direct one-to-one correspondence between $P_{t}$ and $P_{t+1}$, so the scene flow $SF_{t\to t+1}$ represents the motion vector from each point in $P_{t}$ to its closest matching position in $P_{t+1}$. This relationship is modeled by the function *F*:(1)$$SF_{t\to t+1}=F\left(P_{t},P_{t+1},\theta_{t\to t+1}\right),$$
where F(·) denotes the scene flow estimation model and θ represents the model weights. Scene flow estimation is performed for each pair of point cloud frames in the sequence $\left\{P_{0},P_{1},P_{2},\ldots ,P_{N}\right\}$, sequentially calculating the optimal scene flow model weights $\left\{\theta_{0\to 1},\theta_{1\to 2},\ldots ,\theta_{N-1\to N}\right\}$.

For comprehensive multi-frame analysis, the scene flow estimation from any frame *i* to frame *j* is iteratively updated as follows:(2)$$SF_{i\to j}=SF_{i-1\to j}+F\left(P_{i-1},P_{i},\theta_{i-1\to i}\right),$$
where $SF_{i\to j}$ is the cumulative scene flow from the initial frame to frame *j*, refined with the latest estimation of movement between frames i−1 and *i*. This process ensures that each new frame contributes to the overall scene dynamics by updating the cumulative scene flow with the latest available data. Consequently, the scene flow at any given frame *j* is the result of aggregating the motion observed up to that point, thus reflecting a more accurate depiction of the movement patterns over time.

The above formulation ensures that scene flow estimation is treated as an iterative and cumulative process that refines our understanding of the scene dynamics over time rather than defining the point cloud frames themselves.

### 3.2. Two-Branch Pyramid Network

We have developed a two-branch pyramid network to serve as the model for the scene flow estimator, as illustrated in Figure 2. Frame $P_{t}$ is initially inputted into a pyramid feature extractor, which comprises a bottom-up and top-down pathway for the point cloud. The number of sampling layers is configured as 4 for both the bottom-up and top-down pathways, while the scale rate is set to 2. The output of each downsampled layer will be combined with the results of the upsampling layer and passed as input to the subsequent layer for the next upsampling operation. The final output, noted as $P_{t}^{pyr}$, is fed into an implicit regularizer for optimization.

In the implicit regularizer, the backbone is constructed using a neural network based on an MLP. We include a position head and a flow head to encode the positional and motion information of the input point cloud. We adopt the LeakyReLU(Leaky Rectified Linear Unit) [47] as the activation function. Unlike the traditional ReLU activation function, which blocks negative input values, LeakyReLU allows a small, non-zero gradient when the unit is not active. This means that it passes a small fraction of the negative values, thus addressing the issue of ‘dead neurons’ that can occur in ReLU. This characteristic helps in maintaining the flow of gradients through the network, making it particularly useful for models where the preservation of information from negative input values is crucial. The input layer of the backbone consists of three channels, while there are six hidden layers with 128 hidden units each. The backbone is equipped with a 128-channel output, which is subsequently connected to the position head and flow head. Each head comprises two hidden layers, each containing 128 hidden units. The output of each head consists of three channels, which correspond to the estimated coordinates and motion vectors in the XYZ coordinate system. Given the $P_{t}^{pyr}$, the position head outputs the estimated coordinates noted as $P_{t+1}^{\prime }$, and the flow head outputs the estimated flow vectors noted as $SF_{t\to t+1}$.

In the regularizer constraints, we formulate the contraints as two parts. On the one hand, the $P_{t}^{pyr}$ combined with the $SF_{t\to t+1}$ should be consistent with the $P_{t+1}$. On the other hand, the $P_{t+1}^{\prime }$ substracted from the $SF_{t\to t+1}$ should be consistent with $P_{t}$. Thus, the objective function can be defined as
(3)$$\begin{array}{cc} F\left(P_{t}^{pyr},P_{t},P_{t+1},\theta \right)\;=\; & \Psi \left(P_{t}^{pyr}+SF_{t\to t+1},P_{t+1}\right)+\\ \; & \Psi \left(P_{t+1}^{\prime }-SF_{t\to t+1},P_{t}\right), \end{array}$$
where Ψ(.,.) denotes the Chamfer distance function [48], which is defined as
(4)$$\Psi \left(a,B\right)=\min_{b\in B}{∥a-b∥}_{2}^{2},$$
where *a* and *b* are s from point clouds *A* and *B*.

During the optimization process, we follow the gradient descent technique and obtain the optimal scene flow model weights $\theta^{*}$, as shown in Equation (5).
(5)$$\theta^{*}=\underset{\theta }{\arg\min}F\left(P_{t}^{pyr},P_{t},P_{t+1},\theta \right)$$

### 3.3. Kalman Filter Correction

However, although all scene flow model weights {$\theta_{0\to 1}$, $\theta_{1\to 2}$, …, $\theta_{N-1\to N}$} are the optimal estimation of the corresponding point cloud pairs, these parameters tend to suffer from the problem of accumulating localization errors when performing long sequences of scene flow estimation, as depicted in Figure 3.

As depicted in Figure 3, the scene flow estimation model’s performance is visualized. In Figure 3a, we observe that the red point cloud, representing the estimated scene flow, closely aligns with the green point cloud, which represents the ground truth. This alignment indicates a precise estimation without localization error. Conversely, Figure 3b reveals the challenges posed by cumulative localization errors. Despite the estimator maintaining the overall shape of the point cloud, there is a noticeable drift towards the lower left, underscoring the significance of robust error correction mechanisms in long sequence estimations.

To eliminate the cumulative localization error of point cloud during the long sequence estimation process, the Kalman filter [49] is introduced to correct the dynamic targets’ position.

The cluster center of the dynamic target is represented as the state quantity $X={\left[x,y,z,vx,vy,vz\right]}^{T}$, where x,y,z denote the spatial coordinates and vx,vy,vz represent the velocity components. Assuming that the dynamic target adheres to uniform motion, the transition matrix *A* is constructed to reflect this. In the matrix, Δt represents the time interval between successive observations or states. It is used to integrate the effect of time on the movement and velocity of the target. The transition matrix *A* is thus defined as:(6)$$A=\left[\begin{array}{ccccc} 1 & 0 & \Delta t & 0 & 0\\ 0 & 1 & 0 & \Delta t & 0\\ 0 & 0 & 1 & 0 & \Delta t\\ 0 & 0 & 0 & 1 & 0\\ 0 & 0 & 0 & 0 & 1 \end{array}\right]$$

The Kalman filter consists of prediction and updating, as depicted in Figure 4:

$X_{k-1|k-1}$ is the optimal outcome of the previous state, *A* and *B* are the system parameters, $U_{k}$ is the control quantity of the present state, and $X_{k|k-1}$ is the prediction of $X_{k-1|k-1}$. $P_{k-1|k-1}$ is the covariance of $X_{k-1|k-1}$, $A^{\prime }$ is the transpose matrix of *A*, *Q* is the covariance of the system process noise, and $P_{k|k-1}$ is the covariance of $X_{k|k-1}$. *H* is the observation matrix, *R* is the noise covariance of the system measurements, and $Kg_{k}$ is the Kalman gain. $Z_{k}$ is the measured value, $X_{k|k}$ is the optimal predicted value at the current moment, and $P_{k|k}$ is the covariance of $X_{k|k}$.

The center of dynamic target is corrected using the optimal Kalman filter prediction value $X_{k|k}$, which eliminates the localization error frame by frame and obtains accurate scene flow estimation results for a long sequence of point clouds.

### 3.4. Point Cloud Refinement

After accumulating multi-frame point clouds, duplicate points exist in the same part of the target. We propose refining and deduplicating the point cloud to reduce the noise accumulation on the densification result and ensure the uniformity of the target object’s density.

The search radius essentially defines the maximum distance within which a point in the scene flow estimation result $P_{i\to j}$, corrected by the Kalman filter, is considered to be a duplicate of a point in $P_{j}$. The average point cloud distance *D* of point cloud $P_{j}$ is used as the search radius, and the average point cloud distance *D* is calculated by Equation (7):(7)$$D=\frac{1}{n}\sum_{i=1}^{n}\sqrt{{\left(x_{i}-\bar{x}\right)}^{2}+{\left(y_{i}-\bar{y}\right)}^{2}+{\left(z_{i}-\bar{z}\right)}^{2}},$$
where $\bar{x}$, $\bar{y}$, $\bar{z}$ are the mean values of point cloud in *x*, *y*, *z* dimensions, respectively, and *n* is the number of point clouds.

If a point in $P_{j}$ is found within this radius of a point *p*, then *p* is replaced with the nearest point in $P_{j}$; otherwise, *p* is retained. This approach allows us to efficiently deduplicate and refine the point cloud, enhancing the overall quality of the densification results. By iterating this process for all values of *i* ranging from 0 to j−1, we obtain the refined densification results for the first *j* frames.

## 4. Experiments

In this section, we provide a detailed description of the parameter settings used in the scene flow estimation model, along with the hardware and software configuration employed. In the second part, we have finetuned our model on the KITTI scene flow dataset and compared our method against other state-of-the-art scene flow estimation techniques. Next, we proceed to compare the proposed method with the ICP algorithm in terms of the densification outcomes. For the purpose of conducting a comprehensive analysis, we have employed five distinct LiDAR detectors to assess their performance. This evaluation was carried out on both the raw and densified KITTI tracking dataset. The overall methodology and sequential analysis steps in our approach are summarized in Figure 5.

### 4.1. Experimental Setup

The hardware and software configurations utilized in this paper are outlined in Table 1.

In our experimental setup, parallel computing techniques were employed to accelerate the processing, primarily utilizing GPU computation. The workflow involved two key computational stages. Firstly, the scene flow estimation was carried out using GPU-based computations. This step leverages the powerful parallel processing capabilities of the GPU to efficiently handle the complex calculations required by the scene flow model. Following the scene flow estimation, the point cloud positions were corrected using the Kalman filter, a process that was executed on the CPU. For dynamic targets identified in the point cloud space through clustering, multiple CPU threads were allocated to perform Kalman filter operations in parallel. This parallelization on the CPU allowed for efficient handling of multiple dynamic targets simultaneously. Once all the corrections for a particular frame were completed, the process proceeded to the next frame, maintaining this sequence throughout the experiment.

For the task of object detection, our experimental phase primarily focused on the training and validation of the models. This component was conducted separately and predominantly utilized GPU computations. The use of GPUs in this phase was crucial due to their ability to handle the intensive calculations required for deep learning tasks, significantly speeding up the training and validation process of the detection models.

This combination of GPU and CPU parallel processing not only optimized our computational workflow but also ensured the efficient handling of both the scene flow estimation and object detection tasks, crucial for the success of our experiment.

The optimization of the scene flow estimatior is achieved through the utilization of the Adam optimizer, which aims to minimize the objective function. Considering the abundance of point clouds and the issue of overfitting, the learning rate has been set to 1 × 10^−3^, and the optimization rounds have been limited to 5000 iterations.

### 4.2. Scene Flow Results and Analysis

To thoroughly assess our proposed scene flow estimation model, we conducted experiments on the KITTI Scene Flow dataset [50], which comprises 100 training and 50 test scenarios. Each scenario consists of several consecutive stereo image pairs along with corresponding LiDAR point cloud data and precise ground truth annotations, covering diverse driving environments such as urban streets, rural roads, and highways.

The performance of the scene flow estimator was evaluated using a set of established metrics from previous works [51,52]: End-Point Error in 3D (EPE3D), which measures the average Euclidean distance between the estimated and ground truth flow vectors; strict 3D accuracy (acc3d_strict), the proportion of estimated points with an error below 5 cm or 5%; relaxed 3D accuracy (acc3d_relax), considering a threshold of 10 cm or 10%; and the outlier ratio, the percentage of points where the estimation error exceeds 30 cm or 10%. These metrics provide a comprehensive assessment of the estimator’s accuracy and robustness in estimating scene flow, offering a detailed framework for benchmarking against other state-of-the-art techniques. The detailed evaluation metrics are as follows:End-Point Error in 3D (EPE3D): The average Euclidean distance between the estimated flow field ${SF}_{i}^{*}$ and the ground truth ${SF}_{i}^{\mathrm{gt}}$, which provides a measure of the model’s predictive accuracy.Strict 3D Accuracy (Acc3d_strict): The percentage of points with an end-point error of less than 5 cm or a relative end-point error of less than 5%, measuring the model’s accuracy under stringent conditions.Relaxed 3D Accuracy (Acc3d_relax): The percentage of points with an end-point error of less than 10 cm or a relative end-point error of less than 10%, assessing the model’s performance under more lenient conditions.Outlier Ratio (Out.): The percentage of points where the end-point error exceeds 30 cm or the relative end-point error is greater than 10%, identifying predictions with substantial errors.

Following the presentation of evaluation metrics, we now introduce the comparative methods used in our experiments to benchmark our scene flow estimator against current leading techniques in the field, including four distinct scene flow estimation models: Graph Prior [53] employs a graph-based approach for spatial consistency and geometric coherence, beneficial for large-scale datasets in driving contexts. Neural Prior [23] leverages deep learning to encode motion patterns, aiming for robust flow estimation amidst environmental noise and occlusions. SCOOP [54] represents a novel, minimally supervised technique that learns point features and refines flow via soft correspondences, making it suitable for training with limited data. Lastly, OptFlow [55] utilizes an optimization strategy with local correlation and graph constraints, efficiently differentiating between static and dynamic points and showcasing rapid convergence with high accuracy.

The effectiveness of our scene flow estimation model was rigorously evaluated on the KITTI Scene Flow dataset, with the results being compared against several state-of-the-art methods. The comparative results are summarized in Table 2.

Our method achieved the lowest End-Point Error in 3D (EPE3D) of 0.027, indicating the highest precision in estimating the motion vectors. While OptFlow excelled in terms of strict and relaxed 3D accuracy, our method demonstrated competitive performance with a balanced approach towards accuracy (94.3% in Acc3d_strict and 96.8% in Acc3d_relax) and outlier management, evidenced by the lowest outlier ratio of 10.4%. These results signify our method’s capability to accurately estimate scene flow while maintaining robustness against errors. Notably, the non-learning-based nature of our method, alongside Graph Prior, Neural Prior, and OptFlow, reflects the potential of algorithmic approaches without extensive reliance on large training datasets.

To further substantiate the efficacy of our scene flow estimatior, particularly emphasizing the improved localization accuracy provided by the Kalman filter, we include a set of visualization results. Figure 6 concretely demonstrates the scene flow estimation over extended sequences, revealing the trajectories and point cloud data before and after Kalman filter correction. Figure 6a,b illustrate the marked contrast in trajectory prediction and point cloud localization with and without the use of the Kalman filter. In Figure 6a, we observe the corrected trajectories that exhibit high coherence and precision, affirming the filter’s role in enhancing localization accuracy, especially in complex urban and highway scenarios. Conversely, Figure 6b highlights the discrepancies and increased localization errors when the Kalman filter is not employed. The bottom sections of both figures provide a clear side-by-side trajectory comparison, with the corrected paths presenting a compelling case for the Kalman filter’s impact. These visual comparisons not only bolster the quantitative results but also deliver qualitative, intuitive confirmation of our model’s improved performance in dynamic settings. The enhanced flow coherence and diminished error presented in these visual outcomes effectively address the reviewer’s call for more robust visual evidence, thus substantiating our claims of superior scene flow estimation capabilities.

### 4.3. Densification Results and Analysis

The densification experiments are particularly focused on mobile entities like cars, trucks, and cyclists because these entities typically exhibit significant movement and variation in point clouds, presenting unique challenges in terms of densification and accuracy. To disrupt the structure of the raw point cloud, we employ random cropping. Subsequently, we evaluate the performance of our proposed method and the ICP algorithm on these data by conducting a comparative analysis.

In our implementation of the ICP algorithm, the objective was to align the point clouds from consecutive frames. This involved iteratively adjusting the transformation matrix to minimize the distance between corresponding points in the point clouds. The algorithm accounted for rotations and translations to best fit the point clouds from frame to frame. This iterative process continued until the changes in the transformation matrix were below our predefined threshold of 0.005, indicating an adequate alignment had been achieved.

In Figure 7, the process of proposed point cloud densification method is illustrated. Figure 7a shows the original sparse point cloud with missing data sections. Figure 7b demonstrates how the proposed algorithm adds new points to these sections. These new points are generated based on the estimated motion and shape information derived from adjacent frames and the algorithm’s predictive modeling. The points are not imported from other point clouds but are instead synthesized by the algorithm to fill in gaps and enhance the overall density and completeness of the point cloud.

The proposed method and the ICP algorithm are utilized to analyze the dynamic targets that have been segmented from the KITTI tracking dataset. The qualitative comparison results are illustrated in Figure 8. Based on the analysis of the experimental results, it is apparent that our method successfully maintains the shape of the densified point cloud while minimizing the occurrence of noise points.

To evaluate the effectiveness of the method proposed in this correspondence, the evaluation metrics utilized consist of Chamfer Distance (CD) [48] and Root Mean Square Error (RMSE). The Chamfer Distance (CD) is defined by Equation (8). The Root Mean Square Error (RMSE) is defined as
(8)$$RMSE=\sqrt{\frac{1}{N}\sum_{i=1}^{n}{\left(Y_{i}-f\left(x_{i}\right)\right)}^{2}}.$$

The Root Mean Square Error (RMSE) measures the deviation between predicted and true values and is more sensitive to outliers in the data. Based on the above two metrics, a comparative analysis of the performance of the method across different object classes is presented in Table 3.

The Chamfer Distance (CD) measures the consistency of the densification outcome and the ground truth. The Root Mean Square Error (RMSE) is employed to assess the accuracy of the densification result in comparison to the actual values of the inner points. Table 3 demonstrates that the method proposed in this paper achieves lower results for all three classes in both metrics. As shown in Figure 9, we statistically compare the results of densification using one-frame point clouds to twenty-frame point clouds. In most cases, our method achieves better results than the traditional ICP method. The proposed method exhibits superior performance in comparison to the conventional ICP algorithm by effectively improving the density and completeness of the point cloud.

In our ablation studies, we specifically assessed the impact of key components in our proposed method: scene flow, Kalman filter, and point cloud refinement. The analysis of Chamfer Distance results, presented in Table 4, highlights the importance of these components in ensuring spatial accuracy in point cloud densification. Notably, the Kalman filter’s contribution was more pronounced than that of the refinement module; this was especially evident in the results for truck1 and truck2. While the combination of scene flow, Kalman filter, and point cloud refinement provided the most accurate outcomes with the lowest Chamfer Distance across all categories, the Kalman filter alone significantly improved spatial accuracy, underscoring its critical role in our method. The increased Chamfer Distance in configurations without the Kalman filter reaffirms its substantial impact on achieving optimal densification.

In the analysis of Root Mean Square Error (RMSE), detailed in Table 5, we found that the contributions of the Kalman filter and point cloud refinement to the overall precision of densification were similar. The integration of scene flow, Kalman filter, and refinement resulted in the lowest RMSE values, denoting the highest precision. The inclusion of both the Kalman filter and refinement was essential for enhancing localization accuracy and maintaining temporal alignment accuracy in dynamic scenes. This was particularly important for moving objects such as cars and cyclists. The similar performance improvements observed in configurations with either the Kalman filter or refinement alone highlight the balanced contribution of these components to the accuracy of the densification process.

### 4.4. 3D Object Detection Results and Analysis

To evaluate the performance impact of densification on LiDAR-based object detection, we carried out a series of experiments using the KITTI tracking dataset. This dataset was partitioned into a training set with 4001 point cloud frames and a validation set with 3900 frames. The division facilitates a robust evaluation framework, allowing for an in-depth analysis of the detector’s performance under varied conditions.

Densification was applied to the raw KITTI tracking dataset by aggregating data from one and three consecutive point cloud frames. The rationale behind this choice is twofold. Firstly, the original dataset annotations are based on single-frame data, where ground truth labels are provided for each frame in isolation. Introducing more frames for densification would require a complex re-annotation process to include the additional temporal context, a task that would be both labor-intensive and outside the scope of this study. Secondly, the choice of three frames specifically serves to provide a deeper temporal insight while maintaining computational efficiency. It allows us to capture more dynamic changes in the scene than single-frame densification while avoiding the exponential increase in computational load and potential data misalignment that could arise from processing many frames. Our approach, therefore, carefully balances the depth of temporal context with practical constraints such as the availability of labeled data and computational resources.

In our experiments, both one-frame and three-frame densified datasets were used during the training and testing phases to ensure consistency in model evaluation. The training followed the protocol established by the Openpcdet [56] framework, with a designated 200 epochs to achieve detector convergence.

To provide a comprehensive understanding of the detectors used in our study, we briefly describe each:PointPillars [40] is a novel 3D object detection framework for point cloud data. It converts point clouds into a pseudo-image of pillars, enabling efficient detection through 2D convolutional neural networks.SECOND [57] employs sparse convolutional networks to process spatially sparse but information-rich regions in point clouds, thereby enhancing both speed and accuracy in 3D object detection.PointRCNN [58] is a two-stage 3D object detector that operates directly on raw point clouds. It generates 3D bounding box proposals and refines them for precise object localization and classification.PV-RCNN [59] integrates voxel-based and point-based neural networks. This hybrid approach extracts rich contextual features from point clouds, leading to significant improvements in detection accuracy.${\mathrm{Part-A}}^{2}$ [60] focuses on detailed part-level features for robust object recognition in point cloud data. It enhances detection precision by aggregating these fine-grained features effectively.

Our results, as tabulated in Table 6, demonstrate that densification contributes to noticeable performance improvements across these detectors. The PV-RCNN detector, in particular, showcased an enhancement of up to 7.95% in detecting cyclists in challenging scenarios.

However, the performance gains observed with the three-frame densified dataset were marginal compared to those with the single-frame dataset. This outcome suggests that while additional frames do provide more context, there is a threshold beyond which the benefits plateau, likely due to the increased complexity without a corresponding increase in labeled information. In essence, the modest performance gains from additional frames are potentially offset by the introduction of unlabeled objects, leading to a trade-off between temporal context and the quality of training data.

Qualitative comparisons in Figure 10 illustrate the advantages of our densification method. Figure 10b is derived from the PV-RCNN detector trained on the three-frame densified dataset, while Figure 10a pertains to the same detector trained on the raw dataset. Our method enriches the dataset with additional structural and semantic information, leading to a reduction in false positives across various environmental conditions, as highlighted in the image sections. This underscores our densification approach’s ability to mitigate false detections for vehicles, pedestrians, and cyclists.

### 4.5. Limitations and Future Work

Following the detailed presentation of our experimental results and their comparative analysis, it is imperative to acknowledge the limitations of our approach and propose avenues for future advancements.

Limitations:Kalman Filter Integration: Our method uses a Kalman filter to correct localization errors in dynamic targets within point cloud scenes, especially in long sequence scene flows. However, this module’s integration with the deep learning-based scene flow estimator is not as tight as it could be, leading to additional computational overhead.Scene Flow Estimator Constraints: The constraints of our scene flow estimator require further exploration. The model currently faces significant computational challenges when estimating point cloud scene flows in large scenes, indicating a need for further optimization.Validation on 3D Object Detection Models: There is a need for more extensive validation across various 3D object detection models to ensure the method’s adaptability to different applications.

Future Work:Integrating Kalman Filter into Scene Flow Estimation: One avenue for improvement involves embedding the Kalman filter module directly into the scene flow estimation model. This integration, along with developing appropriate constraints, could be advantageous for long sequence point cloud scene flow estimation.Lightweight Data Representation: Adopting more lightweight data representation mechanisms could significantly reduce the overall latency of the algorithm.Experiments with Additional 3D Object Detection Models: Expanding our experiments to include a wider range of 3D object detection models is crucial. Given the issues of multi-frame label mismatch, implementing an appropriate label modification mechanism will be essential to maintain the accuracy of model training.

## 5. Conclusions

This paper proposes a Kalman-based scene flow estimation method for point cloud densification and 3D object detection in dynamic scenes. Our method effectively overcomes the problem of localization errors in estimating long sequence scene flow and improves the accuracy and precision of shape completion. The localization accuracy of the scene flow estimation results can be effectively improved by introducing the Kalman filter to correct the position of the dynamic target. The densification results show that, compared with the ICP method, our method is more suitable for dynamic targets and achieves higher levels of accuracy and precision. Extended experiments on the KITTI 3D tracking dataset prove that our method effectively improves the LiDAR-only detectors’ performance and achieves superior results to the baselines. In future work, we consider optimizing the scene stream estimation speed so as to meet the real-time requirement, and we also consider further fusing the image information to enhance the point cloud to improve the model detection. Our main conclusions in this study can be summarized as follows:With a novel scene flow estimator using a two-branch pyramid network as the implicit regularizer, we are capable of utilizing motion information from multi-frame point clouds to densify point clouds, which achieves excellent performance in both accuracy and precision.The issue of localization error in long-sequence scene flow estimation can be effectively addressed by utilizing a Kalman filter to precisely adjust the position of dynamic objects. This mechanism makes the densification result more robust.Compared with the ICP-based point cloud densification method, our method is based on the implicit constrainer-guided scene flow results, whose ability to fine-tune the position of the point cloud makes the results more accurate and outperforms the traditional methods.Extensive experiments on the KITTI 3D tracking dataset demonstrate that our method effectively improves the LiDAR-only detectors’ performance by adding more semantic and structural details.

## Figures and Tables

**Figure 1 sensors-24-00916-f001:**
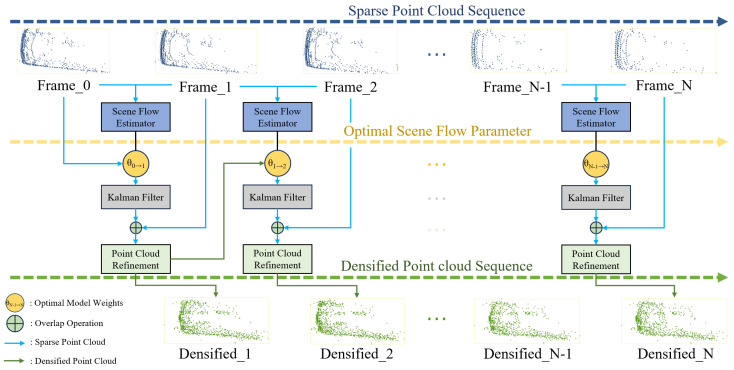
The architecture of densification method. The point clouds are fed into the scene flow estimator to obtain the motion field of the points. Then, the Kalman filter is used to correct the localization error during the long sequence. Finally, the point clouds are refined and deduplicated to obtain the densified results.

**Figure 2 sensors-24-00916-f002:**
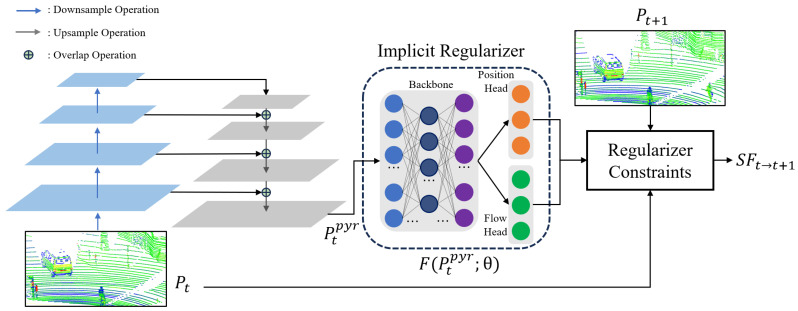
Overall architecture of the proposed two-branch pyramid network. The network mainly consists of a pyramid feature extractor and an MLP-based implicit regularizer. The regularizer constraints are formulated into two parts, which enable the output to satisfy the position and flow consistency. The output $SF_{t\to t+1}$ represents the estimated motion field between $P_{t}$ and $P_{t+1}$.

**Figure 3 sensors-24-00916-f003:**
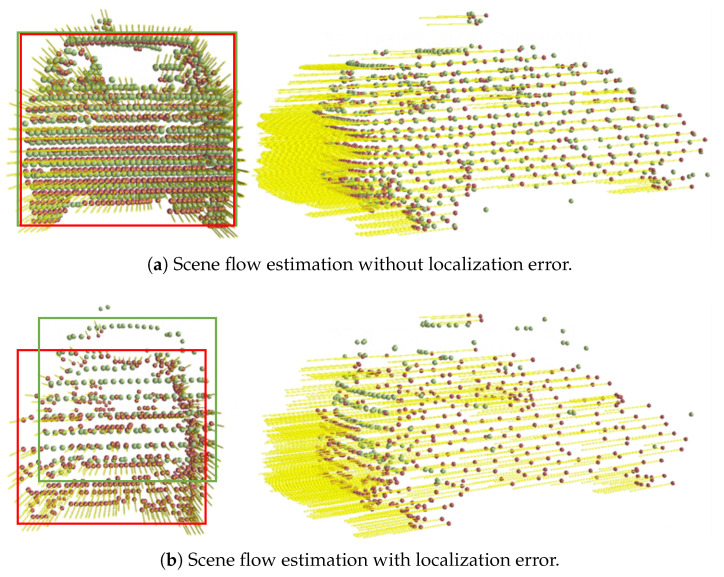
Localization error in long sequence scene flow estimation: (**a**) without localization error, red point cloud matches the green ground truth; (**b**) with localization error, red point cloud shows noticeable drift indicating error significance.

**Figure 4 sensors-24-00916-f004:**
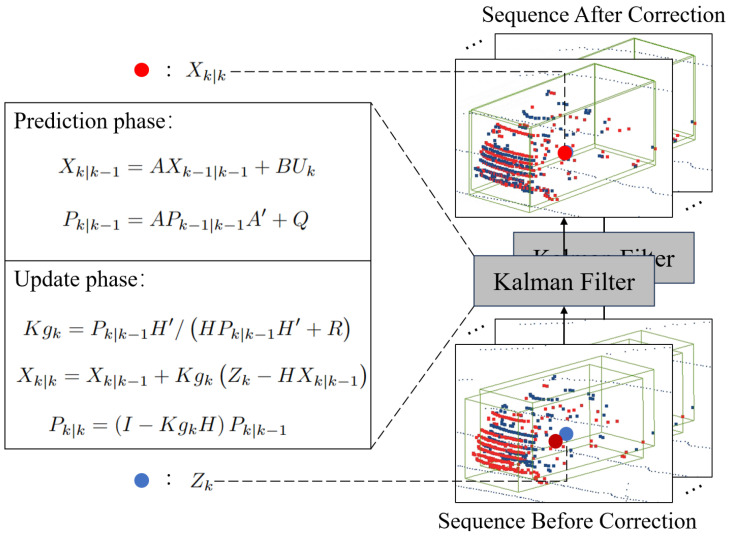
The procedure of Kalman filter correction.

**Figure 5 sensors-24-00916-f005:**
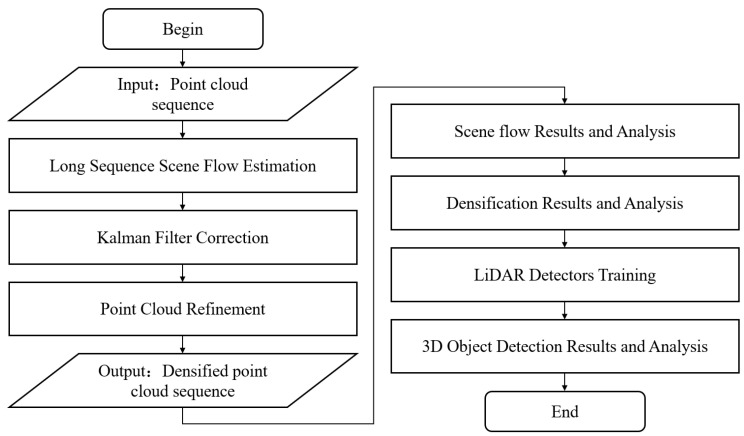
Overall flowchart of the proposed method for point cloud densification and 3D object detection.

**Figure 6 sensors-24-00916-f006:**
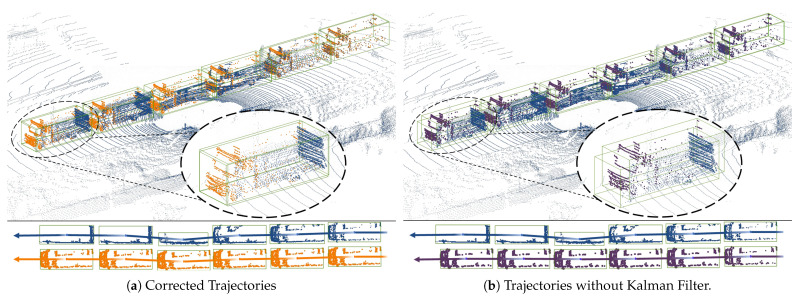
Comparison of scene flow trajectories: (**a**) demonstrates trajectories with Kalman filter correction, with the orange trajectories representing the estimated scene flow after Kalman filter refinement, closely mirroring the blue ground truth trajectories and exhibiting improved localization and flow coherence; (**b**) shows trajectories without Kalman filter correction, where the purple trajectories indicate uncorrected scene flow estimates, leading to visible deviations from the blue ground truth and resulting in increased localization errors. The trajectory comparison at the bottom of the images accentuates the accuracy enhancement due to the Kalman filter, highlighting the precision of our method in tracking and predicting the motion of objects.

**Figure 7 sensors-24-00916-f007:**
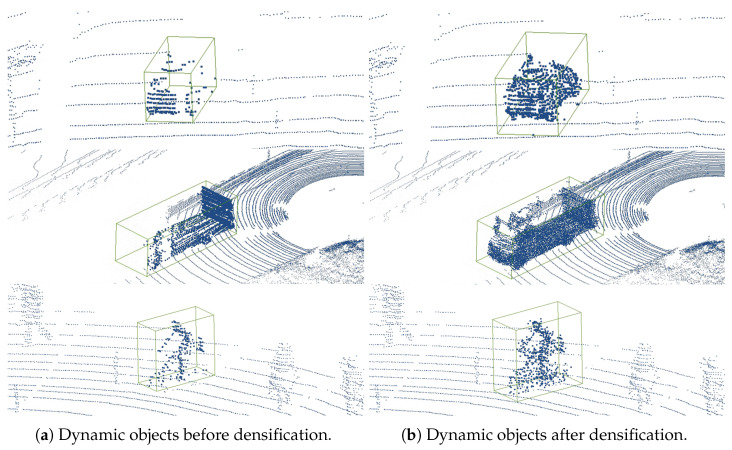
Densification results: (**a**) demonstrates dynamic objects before densification; (**b**) demonstrates dynamic objects after densification. The proposed method effectively provides more semantic and structural information than the raw point cloud.

**Figure 8 sensors-24-00916-f008:**
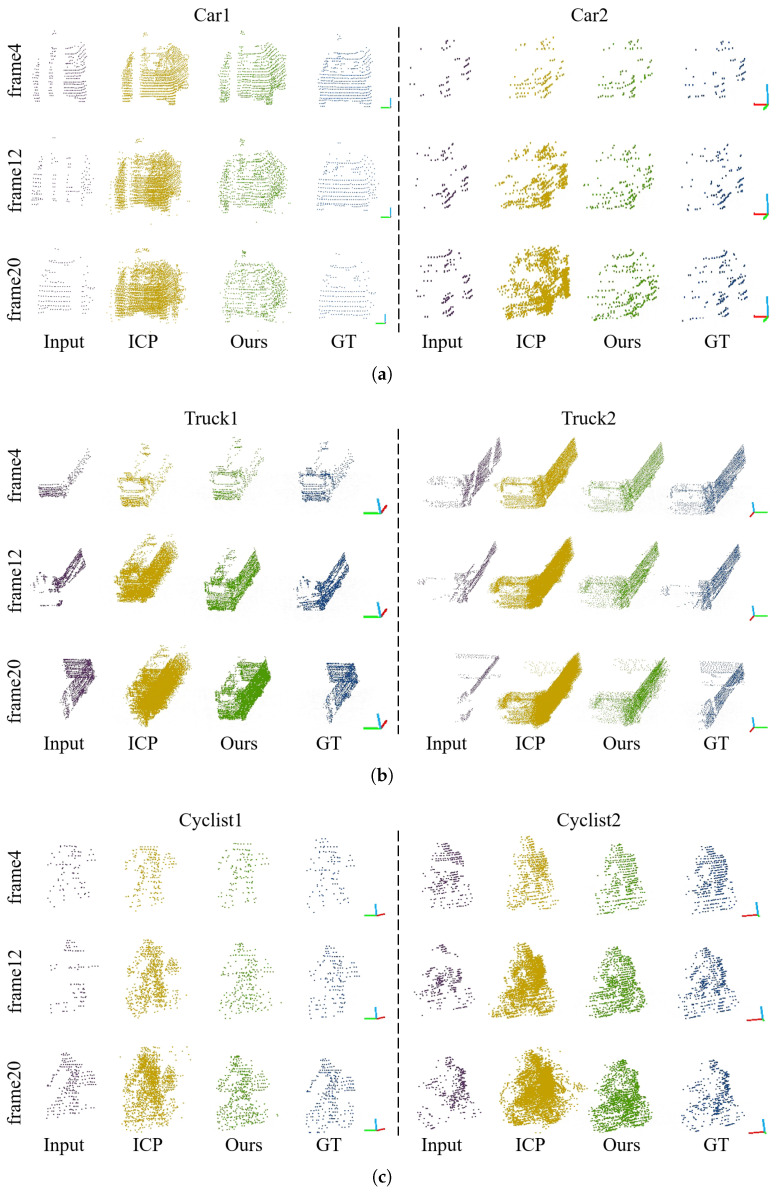
Qualitative comparison of densification results with ICP algorithm: (**a**) cars; (**b**) trucks; and (**c**) cyclists. Densified dynamic objects using our method are depicted in green, and results from ICP algorithm are depicted in yellow. Each object is densified using 4, 12, and 20 frames of the point cloud, respectively.

**Figure 9 sensors-24-00916-f009:**
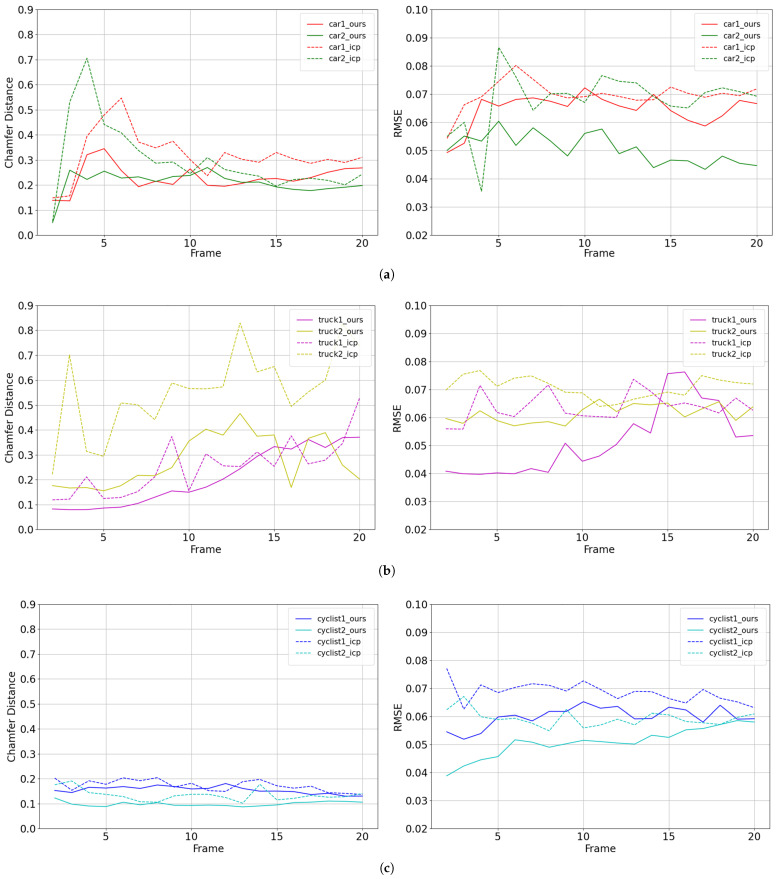
Chart comparison of densification details in Chamfer Distance and Root Mean Square Error: (**a**) cars; (**b**) trucks; and (**c**) cyclists. In most cases, our approach is superior to the ICP algorithm.

**Figure 10 sensors-24-00916-f010:**
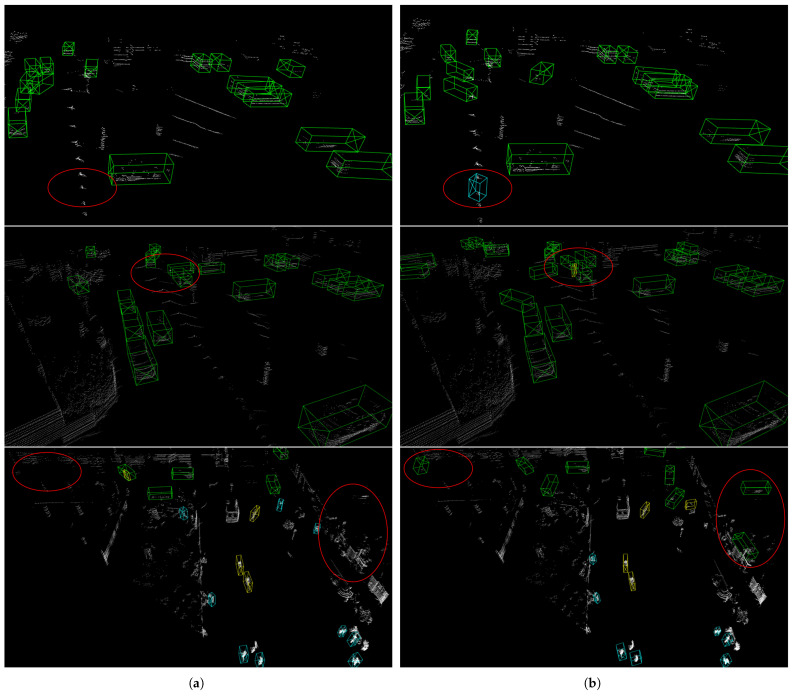
Qualitative comparison of the 3D object detection results: (**a**) the left column shows the results from the PV-RCNN detector trained on the three-frame densified dataset; (**b**) the right column shows the results from the PV-RCNN detector trained on the raw dataset. The boxes in the diagram are green for car, light blue for pedestrain, and yellow for cyclist. Our proposed method effectively reduces the false positive detection in general scenes, as illustrated by the red circles highlighting the detection differences between datasets.

**Table 1 sensors-24-00916-t001:** Hardware and software configurations.

Configuration	Hardware/Software
System	Ubuntu20.04
Environment	Pytorch1.12.0 + CUDA11.6
CPU	Intel i7-12700H@
GPU	NVIDIA RTX 3080 Ti

**Table 2 sensors-24-00916-t002:** Comparative results in the KITTI scene flow dataset.

Method	Learning Based	EPE3D	Acc3d_strict	Acc3d_relax	Out.
Graph Prior [53]	False	0.082	84.0	88.5	-
Neural Prior [23]	False	0.034	92.3	96.4	12.0
${\mathrm{SCOOP}}^{+}$ [54]	Partial	0.039	93.6	96.5	15.2
OptFlow [55]	False	0.028	**96.1**	**97.8**	13.2
Our Method	False	**0.027**	94.3	96.8	**10.4**

Bold indicates the best results.

**Table 3 sensors-24-00916-t003:** Quantitative comparison of point cloud densification performance.

	Car1	Car2	Truck1	Truck2	Cyclist1	Cyclist2
OURS-CD	**0.228993**	**0.206742**	**0.207862**	**0.277185**	**0.155147**	**0.099149**
ICP-CD	0.321194	0.295102	0.250756	0.528042	0.172999	0.134865
OURS-RMSE	**0.064533**	**0.048028**	**0.051484**	**0.061112**	**0.059912**	**0.050888**
ICP-RMSE	0.069794	0.065152	0.063821	0.070297	0.068576	0.059299

Bold indicates the best results.

**Table 4 sensors-24-00916-t004:** Ablation Study Results—Chamfer Distance.

Configuration	Car1	Car2	Truck1	Truck2	Cyclist1	Cyclist2
Scene flow + Kalman filter + Refinement	**0.228993**	**0.206742**	**0.207862**	**0.277185**	**0.155147**	**0.099149**
Scene flow + Kalman filter	0.235468	0.212346	0.231681	0.281356	0.155987	0.100548
Scene flow + Refinement	0.288802	0.249351	0.581330	0.616738	0.239621	0.182909

Bold indicates the best results.

**Table 5 sensors-24-00916-t005:** Ablation Study Results—RMSE.

Configuration	Car1	Car2	Truck1	Truck2	Cyclist1	Cyclist2
Scene flow + Kalman filter + Refinement	**0.064533**	**0.048028**	**0.051484**	**0.061112**	**0.059912**	**0.050888**
Scene flow + Kalman filter	0.065131	0.049756	0.051599	0.061192	0.062988	0.054037
Scene flow + Refinement	0.066378	0.052751	0.054257	0.064125	0.067561	0.058043

Bold indicates the best results.

**Table 6 sensors-24-00916-t006:** Quantitative comparison of five different LiDAR detectors’ performance on raw and densified datasets. Our method achieves superior detection performance in 38 out of 45 categories.

Method	Dataset	mAP	Car			Pedestrain			Cyclist		
		Hard	Easy	Mod.	Hard	Easy	Mod.	Hard	Easy	Mod.	Hard
	Raw Dataset	81.70	97.18	95.49	90.47	66.11	65.48	65.51	92.98	89.61	89.11
PointPillars [40]	One-frame densified	**82.80**	**98.88**	**98.06**	**90.61**	**66.23**	**65.65**	**65.60**	**93.18**	**92.35**	**92.18**
	Three-frame densified	**86.06**	**98.83**	**98.51**	**96.15**	**67.76**	**67.50**	**67.30**	**98.06**	**95.78**	**94.73**
	Raw Dataset	84.24	**98.77**	95.87	90.44	75.02	72.45	72.35	94.61	**94.54**	89.95
SECOND [57]	One-frame densified	**87.34**	98.25	**96.03**	**96.25**	**76.38**	**73.65**	**73.63**	**96.36**	92.21	**92.13**
	Three-frame densified	**87.64**	98.21	**97.42**	**96.15**	**76.15**	**74.28**	**74.69**	**94.78**	92.11	**92.09**
	Raw Dataset	83.93	98.35	90.09	89.90	72.03	73.33	72.98	90.68	90.01	88.92
PointRCNN [58]	One-frame densified	**85.85**	98.19	**90.51**	**90.37**	**82.99**	**77.20**	**77.12**	**90.85**	**90.41**	**90.07**
	Three-frame densified	**86.16**	**98.95**	**90.55**	**90.39**	**80.76**	**79.35**	**77.13**	**96.91**	**91.37**	**90.97**
	Raw Dataset	89.59	99.49	96.70	96.82	**82.34**	**82.05**	**81.91**	98.16	96.17	90.03
PV-RCNN [59]	One-frame densified	**91.54**	**99.50**	**97.33**	**96.89**	81.20	80.85	81.54	**98.78**	96.11	**96.20**
	Three-frame densified	**92.52**	**99.52**	**98.63**	**98.60**	78.49	80.10	80.98	**99.22**	**98.17**	**97.98**
	Raw Dataset	87.09	99.41	90.48	90.35	**81.28**	80.54	**80.86**	98.14	90.18	90.06
${\mathrm{Part-A}}^{2}$ [60]	One-frame densified	**88.85**	**99.48**	**96.26**	**96.51**	80.87	**80.81**	79.94	**98.74**	**95.86**	**90.11**
	Three-frame densified	**89.60**	**96.30**	**94.87**	**95.22**	79.42	78.94	79.44	**98.60**	**94.02**	**94.13**

Bold indicates the best results.

## Data Availability

Data are available in a public repository. The data that support the findings of this study are available in [46] at https://www.cvlibs.net/datasets/kitti/index.php, accessed on 2 October 2023.

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
