# Peer review of "Kalman-Based Scene Flow Estimation for Point Cloud Densification and 3D Object Detection in Dynamic Scenes"

_sensors, 2024, doi:10.3390/s24030916_

Round 1

Reviewer 1 Report

Comments and Suggestions for Authors

The authors propose a Kalman filter-based scene flow estimation method for point cloud densification and 3D object detection in dynamic scenes. The key ideas are using a two-branch pyramid network for scene flow estimation, correcting cumulative localization errors with a Kalman filter during long sequence estimation, and refining the densified point clouds. The method shows improved performance on the KITTI dataset compared to the baseline ICP algorithm and enhances several LiDAR detectors. The paper is well-structured and clearly explains the technical details. The experiments adequately validate the effectiveness of the proposed approach. I have some suggestions to further improve the paper:

1. More comparisons with recent state-of-the-art methods would make the evaluation more convincing. I suggest citing with SSL-Net (PR23), PGF-Net (TIP23), JRA-Net (PR23) and MSA-Net (TIP 22) which also focus on point cloud registration or scene flow estimation. How does the proposed method perform compared to these latest approaches?

2. Ablation studies could provide more insights into the contribution of different components. For example, what is the impact of just using scene flow vs. adding the Kalman filter? How about removing the point cloud refinement step? Some analysis would be useful. 

3. More visualization results demonstrating the improved localization accuracy from the Kalman filter could strengthen the claims. For example, showing trajectories or point clouds before and after Kalman filter correction.

4. The description of the baseline methods for comparison is quite brief. Providing more details on the ICP algorithm and the 5 LiDAR detectors would improve clarity.

5. Some discussion on limitations and future work would add more depth. For example, what are the failure cases or scenarios where the proposed method does not work well? How can the approach be improved or extended in the future?

6. Carefully proofread the paper to fix minor grammar issues, typos, notation inconsistencies, etc. There are a few instances of these throughout.

To summarize, I believe this is a solid contribution with promising results. Addressing the above suggestions can make the paper more convincing for publication. The ideas have merit but require additional validation and discussion to demonstrate novelty compared to state-of-the-art scene flow and point cloud registration techniques. I hope these comments are constructive to improve the work.

Comments on the Quality of English Language

See above

Reviewer 2 Report

Comments and Suggestions for Authors

Please see attached PDF

Comments on the Quality of English Language

Minor edits needed

Reviewer 3 Report

Comments and Suggestions for Authors

The authors proposed a method to improve point cloud data using a scene flow estimator and use kalman filter to correct the localisation error.

Alg 1 and section 3.1 are not consistent, especially for SF

Eq1 does not seem right, the point cloud for the current frame should be independent of the estimated motion flow? It also then does not seem correct with eq 3 with SF for the current frame.

Pt should be defined more clearly in different parts of section 3, is it the raw point cloud, the point cloud after the scene flow estimator or a frame?

In fig1, the point cloud from frame 0 and 1 are inputs to the estimator but in fig 2, it seems only one frame is the input?

Section 4.3 is not clear whether the densification is applied only during testing or in training too.

In section 4, it is unclear why 1 frame and 3 frame are chosen for densification.

Experimental results seems convincing but it would be interested to test on other public datasets such as scannet or sun-rgbd.

Comments on the Quality of English Language

The paper is well-structured but is difficult to read in some sections. The section 2 could be rephrased without having the references as sentences subjects. There is a lack of information that makes it unclear. It would be easier to read and understand.

There is a typo in Alg1 step5 and F function is not defined in Alg. 1

References formatting is not consistent.

Round 2

Reviewer 1 Report

Comments and Suggestions for Authors

We think the authors have carefully considered our problems. This version is fine for me.

Comments on the Quality of English Language

See above